# Evaluation of the Level of Technical and Tactical Skills and Its Relationships with Aerobic Capacity and Special Fitness in Elite Ju-Jitsu Athletes

**DOI:** 10.3390/ijerph182312286

**Published:** 2021-11-23

**Authors:** Tadeusz Ambroży, Łukasz Rydzik, Michał Spieszny, Wiesław Chwała, Jarosław Jaszczur-Nowicki, Małgorzata Jekiełek, Karol Görner, Andrzej Ostrowski, Wojciech J. Cynarski

**Affiliations:** 1Institute of Sports Sciences, University of Physical Education in Krakow, 31-541 Kraków, Poland; michal.spieszny@awf.krakow.pl (M.S.); wieslaw.chwala@awf.krakow.pl (W.C.); andrzejostrowski@poczta.fm (A.O.); 2Department of Tourism, Recreation and Ecology, University of Warmia and Mazury, 10-719 Olsztyn, Poland; j.jaszczur-nowicki@uwm.edu.pl; 3Department of Ergonomics and Physiological Effort, Institute of Physiotherapy, Jagiellonian University Collegium Medicum, 31-126 Krakow, Poland; malgorzata.jekielek@gmail.com; 4Department of Physical Education and Sports, Matej Bel University in Banská, 974-01 Banská Bystrica, Slovakia; gornerk@uek.krakow.pl; 5Institute of Physical Culture Studies, College of Medical Sciences, University of Rzeszow, 35-959 Rzeszów, Poland; ela_cyn@wp.pl

**Keywords:** physical fitness, training control: special fitness test, martial arts

## Abstract

Background: Ju-jitsu training has to be comprehensive in terms of training intensity, developing a wide range of physical fitness and learning multiple technical skills. These requirements result from the specificity of the competition characteristic of the sport form of this martial art. The aim of this study was to evaluate the aerobic capacity and special physical fitness of ju-jitsu athletes at the highest sports performance level and to determine the relationships between special fitness and the indices of technical and tactical skills. Methods: In order to determine the current level of special fitness of the athletes, a set of karate fitness tests were used, namely, the Special Judo Fitness Test and the Kickboxer Special Physical Fitness Test. Furthermore, maximal oxygen uptake (VO_2_peak) was measured using a graded exercise test in a group of 30 sport ju-jitsu athletes at the highest level of sports performance. To evaluate the level of technical and tactical skills, an analysis of recordings of tournament bouts was carried out, and, based on the observations, the indices of effectiveness, efficiency, and activeness of the attack were calculated. Results: Individuals with higher fitness were more active and effective in the attack. The special efficiency indices showed significant correlations with the technical and tactical parameters. Better fighting performance was dependent on the speed of the punches, kicking range, and the results of the special fitness tests. Conclusions: To achieve greater efficiency and effectiveness of sport ju-jitsu, the training process should be based on comprehensive motor development and an optimal level of special fitness.

## 1. Introduction

In Poland, modern sport ju-jitsu involves extensive competition in the form of tournaments organized by the Polish Ju-jitsu Association in the following variants: Fighting System, Duo System, and Ne Waza [1]. This makes it necessary to constantly improve physical abilities and to expand the range of techniques used by athletes during sports combat. With such an extensive formula of competition, the training process in sports ju-jitsu must be versatile in terms of training intensity, developing a wide range of motor skills, and mastering multiple technical skills. A review of the literature indicates that during the observation of the course of fights during the Junior Ju-jitsu World Championships (Bucharest 2013) the most frequently used hand techniques in the first phase of the bout were punches and inverted fist techniques, while the kicks included side kick, roundhouse kick, and front kick. In the second phase of the bout (characterized by grappling), hand throws (morote gari and seoi nage), sweep techniques (osoto gari, osoto otoshi, and ouchi gari), and hip throws (goshi guruma and harai goshi) were used most often, while in the third phase, pinning techniques, joint locking, and choking techniques were most frequent [2]. The analysis of the course of the fight conducted by other authors [3,4] suggests that the ju-jitsu bout in the Fighting System resembles karate or kickboxing in the first phase, while in the second and third phase, the fight is similar to judo [5,6]. In sport ju-jitsu, the fight is characterized by acyclic work, and the changing situation requires the athlete to control the situation and respond quickly to the opponent’s actions. Physical effort during combat in sport ju-jitsu is based on submaximal and maximal training load [7]. For this reason, energy source is anaerobic glycolysis, while aerobic sources are utilized at the end of the bout. The presented literature analysis indicates that optimal aerobic and anaerobic endurance is necessary in ju-jitsu training [8]. An important objective of the ju-jitsu training is to achieve a high level of anaerobic power (dynamic kicks and throws), strength (throws and ground holds), and limb speed (punches and combinations in the attack, blocks, and dodges in defense) [3,9,10,11,12,13]. The relationships between physical fitness and tournament bouts in ju-jitsu were also established by evaluation of fitness using general and special tests. The research focused on sport ju-jitsu, Brazilian ju-jitsu, and judo [14,15].

Fighting in ju-jitsu is characterized by the use of a wide range of techniques combined into specific sequences. An appropriate level of technical and tactical skills seems to be the most important element of an athlete’s success. Technical and tactical actions are used to control the fight and respond to the opponent’s attacks, while preventing counterattacks [3,16,17,18]. As part of the coach’s supervision, the analysis of combats is used by determining the indices of the athlete’s technical and tactical skills, which allows for the assessment of their potential competitive performance. Such an analysis is a popular indicator used to modify the sports training in martial arts and combat sports. These issues have been studied in detail in judo [19,20,21] and kickboxing [22,23]. The review of the literature on ju-jitsu shows that, to date, studies have mainly focused on the evaluation of post-training adaptations and a level of the athlete’s physical fitness [13,14,24,25,26,27], and analysis of morphological [28,29], physiological [30], and biomechanical indices [31,32]. Other studies have focused on the analysis of the ju-jitsu technique [33], the required profiles in ju-jitsu [34,35], the structure of the fight [36], health parameters [29], optimism, and satisfaction with life [37].

A detailed literature review revealed that there is a lack of research on the analysis of the level of technical and tactical skills, especially concerning special physical fitness.

The aim of this study was to evaluate the aerobic capacity and special physical fitness of ju-jitsu athletes at the highest sports performance and to determine the relationships between special fitness and the indices of technical and tactical skills. Evaluation of the correlations will demonstrate whether the level of special physical fitness determines the activeness, effectiveness, and efficiency of the attack and will enable more effective planning of sports training.

## 2. Materials and Methods

The study was conducted on a group of 30 sport ju-jitsu athletes with the highest level of sports performance who agreed to participate in the study. The selection of the study group was purposeful and the selection criterion was training experience (at least 5 years) and level sports performance (champion level), which was assessed based on the authors’ observations and the opinion of coaches. The athletes were included in the ranking list of the Ju-Jitsu International Federation and the Polish Ju-jitsu Association, and four of them were medalists of the world championships. The mean participants’ experience in sport was 7.1 ± 2.26 years. They trained 1.5 to 2 h, 6–10 times a week. The exclusion criterion was a negative opinion of a sports physician and no consent to participate in the study. The mean age of the respondents was 25.43 ± 1.96 years, body weight was 83.74 ± 8.6 kg, and body height was 180.96 ± 4.97 cm. BMI ranged from 21.4 to 29.1 kg/m^2^. Body mass was measured using a Tanita BC-601 body composition monitor (Tanita, Tokyo, Japan), whereas the body height was measured using a SECA 2017 body height meter (Seca, Hamburg, Germany).

### 2.1. Measurement of Physical Fitness and Aerobic Capacity

Aerobic capacity was measured using the aerobic maximal graded treadmill test [38]. The test evaluated maximal oxygen uptake (VO_2_peak) expressed in milliliters of oxygen per kilogram of body weight per minute (ml/kg/min). During the test, the levels of cardiopulmonary indices were recorded based on the breath-by-breath method using an ergospirometer (Cosmed, Rome, Italy). The test was performed on a treadmill (Saturn 250/100R,h/p/Cosmos, Munich, Germany). The effort began with a 4 min warm-up performed at a speed of 8 km·h ^−1^, with a treadmill inclination angle of 1°. Thereafter, the speed was incremented by 1.0 km*h^−1^ every 2 mins. When the heart rate (bpm) approached the maximum level, the running speed was maintained and the load was increased by changing the angle of the treadmill by 1° every minute. The test was carried out until the participant refused to continue the test due to volitional exhaustion. The heart rate (HR) during the test was measured with a sports tester (S-610i, Polar, Finland). The following indices were analyzed: pulmonary ventilation (VE), oxygen uptake (VO_2_), carbon dioxide production (VCO_2_), respiratory-exchange-ratio (RER), expiratory carbon dioxide concentration (%FECO_2_), the ventilatory equivalent ratio for oxygen and carbon dioxide (VE/VCO_2_), and heart rate (HR). Data were averaged every 30 s. The highest recorded value of oxygen uptake was considered as peak oxygen uptake [39].

Special physical fitness was assessed based on selected tests taken from special fitness tests in karate, kickboxing, and judo:Evasive action test: In the retreat test, the subject starts from the fighting stance, moving backwards between lines 8 m apart. The track in the shape of a loop between the lines (back and forth) is covered 6 times. Execution time is measured in seconds. Equipment: stopwatch and 2 small cones that are set on lines at a distance of 8 m [40].Hip turning speed test: In the test of the speed (frequency) of the hip turns, each subject should be tied with a belt over his or her right hip (unless they usually fight in the reverse position), take a fighting stance, and twists the hips to the left. This movement will tighten the belt held by the coach at the back (control).Then the subject retracts with his or her hips. On the command “Hajime!” (“Forward!”), the participant makes 30 hip turns as quickly as possible (belt pulls are counted). The time taken to execute 30 pulls is recorded. Equipment: stopwatch and ju-jitsu belt [40].Speed punches test: Punches by the fighter performed from the fighting stance. Each participant performs a combination of two punches: left, straight to the head (Oi seiken jodan tsuki) and right, straight to the torso (Gyaku seiken chudan tsuki), without changing the designated distance. The targets to which the participant performs 30 such combinations (60 punches in total) are held by a second person at a constant height. The time to complete 30 full punches is measured in seconds with an accuracy of 0.1 s. Two assistants need a stopwatch and 2 targets [40].Flexibility test using Mawashi Geri kick: The flexibility index = maximum range of kick/body height (cm/cm) was calculated. In the flexibility test, the maximum range (foot hit height) of the Jodan Mawashi Geri roundhouse kick was evaluated. Five measurements were taken for the preferred limb by recording the maximum score (cm) [40].Special Judo Fitness Test (SJFT):

Before the first test, the participant performs a warm-up consisting of 5 min of jogging (moderate intensity) and a few Ippon-seoi-nage throws at a slow pace to familiarize themselves with the distance and exercise partners [41,42].

This test consists of three periods of effort (A = 15 s; B and C after 30 s) separated by 10 s breaks. In each period/series of throws, the thrower’s (tori) score is based on the maximum number of Ippon-seoi-nage throws on two partners (uke A and B) standing 6 m apart on the mat. Both uke A and B should have a similar body height and weight to the tori. Heart rate is measured one minute after the test. The Index in SJFT was also counted as
$$SJFT\;Index=\frac{Final\;HR\left(bpm\right)\;+\;HR1\;min\;\left(bpm\right)}{Throws\;\left(N\right)}$$
where:

*Final HR*—heart rate recorded immediately after the test*HR1 min*—heart rate recorded 1 min after the test*Throws*—number of throws completed during the test

The body’s response to exercise was recorded using the heart rate monitor S-610i (Polar, Finland).

6.Kickboxer Special Physical Fitness Test (KSPFT):

Before performing the test, the participant performs a warm-up consisting of approximately 10 min of a general warm-up, followed by stretching exercises. Then the participant performs left and right straight punches on the partner’s shield from the fighting position to the head continuously for 30 s. After completing this part of the test, the participant runs 10 m in a straight line to the next station, where they perform left and right roundhouse kicks from the fighting position to the partner’s shields for 30 s to the head level. Then the participant runs back to the first shield and performs techniques for 30 s in a sequence consisting of the left straight punch, followed by the right hook to the head. After completing this part of the test, the participant runs 10 m to the target mate and performs a roundhouse kick (left and right feet, alternately) to the torso for 30 s. The total time of special effort during the test is 2 min (4 × 30 s), which corresponds to the duration of one round of a kickboxing bout. Figure 1 shows a diagram of individual test components and the direction of the athlete’s movement. Correctly performed hits are counted for each of the four parts. The sum of strikes (punches and kicks) is recorded as the score [43,44].

The tests were supervised by the authors of the present study for 3 days, two weeks before the competition, after the completion of the direct pre-competition training period. On the first day, the maximal graded exercise test was carried out. On the second day, the special fitness tests (tests 1–5 given above) were performed, whereas the KSPFT test was scheduled for the third day. Two days before the test, the training intensity was reduced to 30–40%.

### 2.2. Measurement of Indices of Technical and Tactical Skills

Analysis of bouts was performed based on digital recordings of selected tournament fights of the athletes in 2020. Three fights of each athlete were analyzed and semifinal, or final fights, were taken into consideration. The recordings were made using three cameras (Sony HDR-CX115, Manufacturer, Tokyo, Japan). The video editing program Movavi Video Editor 14 was used to process the images. The setting of the cameras allowed for continuous observation of the fighting athletes, judges, and the scoreboard. A single spreadsheet was developed as a primary research tool. The data from the spreadsheets were entered into Excel software. Then, the values of indices of technical and tactical skills were calculated for all three stages of the bout. The formulas for calculating the indices were developed based on the formulas used for judo and kickboxing [23,45].


**Efficiency of the attack (S_a_)**

$$S_{a}\;=\frac{\left(n1\;x\;1\right)\;+\;\left(n2\;x\;2\right)\;+\;\left(\;n3\;x\;3\;\right)}{N}\;$$



n1—number of attacks assessed in waza-ari as 1 pointn2—number of attacks assessed in ippon as 2 pointsn3—number of attacks assessed in ippon as 3 points1,2,3—point values of successful attacksN—sum of observed bouts


**Effectiveness of the attack (E_a_)**

$$E_{a}\;=\frac{number\;of\;effective\;attacks}{total\;number\;of\;attacks}\;\times \;100$$



*An effective attack is a technical action awarded a point*Number of all attacks is the number of all offensive actions


**Activeness of the attack (A_a_)**

(1)
$$A_{a}\;=\frac{number\;of\;attacks\;recorded\;for\;the\;athlete}{number\;of\;bouts\;performed\;by\;the\;athlete}\;$$



### 2.3. Bioethics Committee

Prior to participation in the tests, the participants were informed about the research procedures, which were in accordance with the ethical principles of the Declaration of Helsinki WMADH (2000). The participant’s written consent was the inclusion criterion. The research was approved by the Bioethics Committee at the Regional Medical Chamber (No. 287/KBL/OIL/2020).

### 2.4. Statistical Analysis

The statistical analysis of the collected material was performed using the Statistica 13.1 package by StatSoft. Basic descriptive statistics were calculated: arithmetic means, 95% confidence intervals, median, minimum and maximum, first and third quartiles, and standard deviations. The consistency of the distribution with the normal distribution was verified using the Shapiro–Wilk test, whereas the relationships between individual variables were calculated using the Pearson linear correlation. The following ranges of correlation coefficient r were adopted: low up to 0.2, mild for 0.2 to 0.5, moderate for 0.5 to 0.8, and high for >0.8. The level of statistical significance was set at *p* < 0.05 [46].

## 3. Results

Table 1 and Table 2 present the means, standard deviations, medians, and basic statistical analysis of the physical capacity and special performance parameters of the ju-jitsu athletes studied. Table 3 presents the indices of activeness, efficiency, and effectiveness of the attack. Table 4 shows the statistical relationships between the indices of technical and tactical skills and the physical fitness of the athletes. Aerobic capacity results ranged from 42.29 to 59.21 mL/kg/min, with a mean of 51.83 mL/kg/min (Table 1).

The activeness of the attack was calculated at a mean level of 15.49 and ranged from 13.11 to 41.12. The mean effectiveness s of the attack was 48.93, and it ranged from 45.12 to 51.23. The efficiency of the attack was at a mean level of 18.15, ranging from 16.32 to 19.82 (Table 3).

People with a greater kicking range were more active (r = 0.55), efficient (r = 0.67), and effective in the attack (r = 0.73) (moderate correlation). In all three categories of the indices, higher values were achieved in the special throwing fitness test in terms of the number of completed throws. The activeness and effectiveness of the attack correlated significantly with the speed of punches. The activeness and efficiency of the attack depend on the accuracy, precision, and speed of kicks and punches, which is illustrated by the significant moderate correlations between these indices and the Kickboxer Special Physical Fitness Test. The activeness, effectiveness, and efficiency of athletes expressed by the indices of technical and tactical skills showed a strong dependence on the level of maximum oxygen uptake (VO_2_peak) (Table 4).

There were no statistically significant correlations between the maximal oxygen uptake (VO_2_peak) and the results of special fitness tests (Table 5).

## 4. Discussion

In the present study, we performed an innovative analysis of the bouts in sport ju-jitsu by measuring indices of technical and tactical skills and determining their correlations with special fitness. Based on the literature review concerning the relationships of physical fitness with technical patterns in sport ju-jitsu, the correlations occur with a high-level of hand-eye coordination, maximum anaerobic power, arm strength, endurance, speed of movement, and flexibility [13,25,47,48]. The results of special fitness tests of athletes studied were similar to those achieved by elite ju-jitsu athletes in previous studies [13]. When competing at the elite level, the appropriate aerobic capacity is important [30,49]. In this study, the mean VO_2_peak reached the value of 51.83 ± 3.37 mL/kg/min, which is considered a high level [8,50,51]. In other similar sports, the average level of maximum minute oxygen uptake in Polish athletes was 40.8 mL/kg/min (judo athletes), 50.3 mL/kg/min (boxers), 58.4 mL/kg/min (MMA fighters), and 47.65 mL/kg/min (kickboxers) [22,52,53,54]. Therefore, the athletes were characterized by a level of aerobic capacity similar to judo athletes, kickboxers, boxers, and MMA athletes. The technical structure of MMA combats is similar to those in ju-jitsu [55], but their duration is much longer, ranging from 15 to 25 min. Therefore, a higher level of aerobic endurance is needed. Other studies have indicated that the mean VO_2_peak in elite kickboxers ranged from 54 to 69 mL/kg/min [56,57], while in the judo athletes, it was 53.75 mL/kg/min [58]. Comparison of the results of our research to these findings reveals that in the present study, ju-jitsu athletes had a lower level of aerobic capacity than the world’s elite kickboxers and judo athletes. However, it seems that aerobic capacity is an important element of the physiological fitness of ju-jitsu athletes, which results from the course of the fight. It is worth noting that the bout in sport ju-jitsu consists of periods of intense exercise in standing position, ground fighting, and breaks. The average time of a bout with breaks is 193.6 s [3]. Given the duration of the bout, it can be assumed that the athletes use aerobic metabolism. This analysis shows that aerobic capacity is essential for achieving the optimal level of special strength. Analysis of combats seems to be the best tool for a reliable assessment of a competitive activity of an athlete. For this reason, special fitness tests are used in combat sports, with their contents corresponding to the temporal and material structure of the fight [13,44]. In this study, we used the most effective tests for assessing the special fitness of athletes based on elements of judo and kickboxing. The results of our research showed differences between the ju-jitsu athletes and judo athletes in the results of the special throwing test of the SJFT (index) in favor of judo practitioners [59]. As the SJFT provides comprehensive information on the performance requirements necessary to take up a fight [60], the development of special throwing endurance can also be used to improve the activeness of the sport ju-jitsu athletes during tournaments. In terms of the total number of kicks and punches performed during the KSPFT test (288.68 ± 31.51), the athletes studied here were better than kickboxers (275.3 ± 31.1), karatekas (271.5 ± 47.7), and non-elite ju-jitsu athletes (269.1 ± 47.2) [43]. These results indicate an adequate level of technical preparation for tournament efforts during the first stage of the bout. The KSPFT used in our research allows for the assessment of the level of technical skills in athletes in terms of the most frequently used upper limb strikes (punches) and lower limb strikes (kicks), speed (number of punches and kicks per unit of time), and special endurance (number of strikes), coordination (combination of strikes), and flexibility (kicking range) [44]. The most important factors influencing the success include an appropriate level of technical and tactical skills and the potential and ability to use them effectively during a tournament fight. The level of technical-tactical skills has an impact on the course of the fight. Therefore, when describing a fight in sport ju-jitsu, one should bear in mind such elements as actual fighting time, work and rest time, and number and types of technical and tactical activities performed by athletes (activeness, efficiency, and effectiveness of the attack) [4,61]. Performing actions in both attack and defense affects the final result of the fight and helps the athlete score points [62]. When planning an optimal training program to ensure that the athlete reaches the champion level, the information obtained from the monitoring of the fights that determines the so-called ju-jitsu combat model should be used. The most popular form of the evaluation of sports performance of an athlete are the indices of technical and tactical skills, which can be used by coaches to adjust training programs and determine the current level of skills for an athlete [61].

The most important aspect of this study was to demonstrate the potential relationships of the indices of technical and tactical skills and physiological parameters with the general and special fitness of ju-jitsu athletes. To control the sports combats in ju-jitsu, a detailed analysis of the temporal and material structure was performed [3,4]. In this study, indices of technical and tactical skills were used for the first time to characterize the athletes’ performance. They were illustrated using the data collected during the ju-jitsu tournaments, calculated according to the author’s formulas, and prepared based on well-established and popular indices used to evaluate the performance of judo [45,63] and kickboxing athletes [23].

Ju-jitsu athletes with greater kicking ranges were more active, efficient, and effective in the attack, as indicated by high positive correlations of indices of activeness, efficiency, and effectiveness of the attack with the flexibility index. Athletes fighting in the Fighting System variant of sport ju-jitsu often use kicks to the body trunk and head in the first phase of the fight [4]. These actions make it possible to keep the opponent at an appropriate distance without giving them the opportunity to attack. This way of fighting may indicate the need for a high-level of flexibility, which may also, in some situations, partially compensate for the poorer performance in the second and third phases of the bout. It is also worth noting that Ambroży et al. demonstrated that the high roundhouse kick is the most effective leg technique [64]. No significant correlations were found between VO_2_peak and special fitness tests. This may be due to the fact that the duration of the tests is relatively short and takes place at the level of anaerobic exercise, while the duration of the entire bout, especially during tournaments, requires optimal levels of aerobic endurance [4].

One of the most important results of the present research is the assessment of correlations between special fitness test scores and activeness, efficiency, and effectiveness of attack. The results showed positive correlations between the three indices and the results in the special throwing fitness test. Therefore, it can be concluded that the throwing skills used in the second phase of the bout determine the effectiveness of the attack, which is proved by the positive correlations in the number of throws completed in the test, in relation to the indices of technical and tactical skills. The positive correlation between SJFT (Index) and the indices of technical and tactical skills can be explained by the fact of a high heart rate observed in the athletes, which could be genetically or emotionally determined, and affected the final value of the index. It is also worth identifying other elements related to the level of technical and tactical skills, e.g., searching for the dependency of special efficiency on individual techniques and determining the value of the elements that were not examined (overthrows and grappling).

The activeness and efficiency of the attack also depend on the accuracy, precision, and speed of kicks and punches, which is illustrated by the significant correlations between these indices and the Kickboxer Special Physical Fitness Test (KSPFT). According to Ouergui et al., a high level of combat activity directly translates into its result fight [65]. The results of our research clearly indicate that the most important element of the high level of technical and tactical skills of sport ju-jitsu athletes (Fighting System) during tournaments is an adequate level of special fitness (especially endurance and special speed) and technical skills (in terms of all striking techniques and throws) used in the first, second, and third phases of the bout.

## 5. Conclusions

The activeness, effectiveness, and efficiency of the attack, expressed in terms of the level of the athlete’s technical and tactical skills, show a dependence on the level of maximum oxygen uptake (VO_2_peak), in the range of r = 0.57–0.59.

It follows that ju-jitsu athletes should develop in the preparation period, and then maintain in the competitive season, an optimal level of aerobic capacity, which should have a direct effect on their competitive performance.

The most important element of the high level of technical and tactical skills in sport ju-jitsu athletes competing in the Fighting System variant is an adequate level of special fitness (especially endurance) and technical skills (in terms of all striking techniques and throws) demonstrated in all combat phases.

### 5.1. Practical Implication

The training programs for ju-jitsu athletes competing in the Fighting System variant during tournaments should be focused on the development of an optimal level of special fitness. Improvements in the competitive performance of ju-jitsu athletes can be achieved by increasing the level of special endurance and quality of technique in performing throws, kicks, and strikes.

### 5.2. Limitation of the Study

Only the results of fitness tests based on the same testing methodology as in the present study were analyzed as comparative material. The literature review was made only in the field of the sport studied, i.e., sport ju-jitsu (Fighting System). Other forms of sport ju-jitsu (Duo, Brazilian ju-jitsu, Ne-Waza) were not taken into account.

## Figures and Tables

**Figure 1 ijerph-18-12286-f001:**
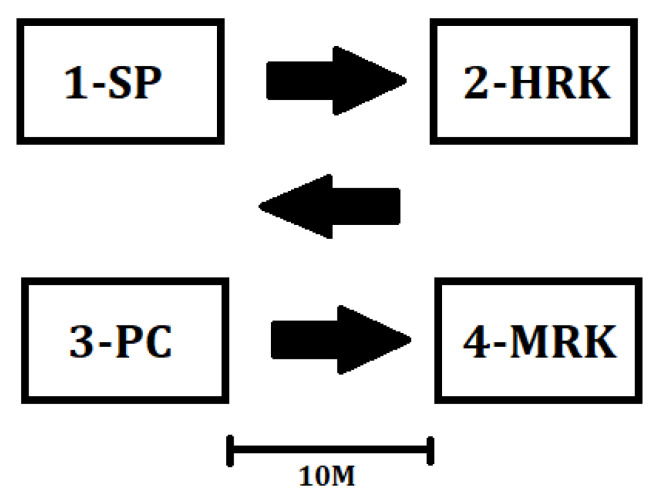
Graphical diagram of the Kickboxer Special Physical Fitness Test. Source: own study. SP—straight punches/jabs, punches; HRK—high roundhouse kick; PC—a combination of straight punches and hooks (punch combinations); MRK—middle roundhouse kick.

**Table 1 ijerph-18-12286-t001:** Fitness level of athletes studied expressed by VO_2_peak and maximum heart rate.

Variables	No	m	−95%CI	+95%CI	Me	Min	Max	Q1	Q3	SD
VO_2_peak [ml/kg/min]	30	51.83	50.54	53.11	51.05	45.29	59.21	49.63	55.32	3.37
HR max [bpm]	30	190.48	187.99	192.96	189.00	180.00	208.00	186.00	195.00	6.53

No—number, m—mean, 95%CI—confidence interval, Me—median, Min—minimum, Max—maximum, Q1—first quartile, Q3—third quartile, SD—standard deviation.

**Table 2 ijerph-18-12286-t002:** Special physical fitness of the respondents.

Variables	No	m	−95%CI	+95%CI	Me	Min	Max	Q1	Q3	SD
Evasive action test [s]	30	42.74	42.26	43.22	42.60	40.33	45.95	41.80	43.29	1.25
Hip turning speed test [s]	30	11.97	11.58	12.35	11.76	10.06	14.15	11.40	12.70	1.00
Speed punches test [s]	30	11.83	11.42	12.25	11.82	10.02	13.90	11.03	12.71	1.08
1/Flexibility Ind	30	0.99	0.96	1.03	0.98	0.87	1.32	0.94	1.02	0.09
SJFT	30	14.56	14.18	14.94	14.63	12.59	16.61	14.02	15.24	1.00
KSPFT	30	288.68	276.70	300.67	290.00	239.00	349.00	263.00	312.00	31.51

1/Flexibility Ind—flexibility index for the test using Mawashi Geri kick [m] SJFT—Special Judo Fitness Test, KSPFT—Kickboxer Special Physical Fitness Test.

**Table 3 ijerph-18-12286-t003:** Activeness, efficiency, and effectiveness of the attack in the group of athletes studied.

Variables	No	m	−95%CI	+95%CI	Me	Min	Max	Q1	Q3	SD
Activeness	30	15.49	13.60	17.38	14.60	13.11	41.12	14.20	15.09	4.97
Effectiveness	30	48.93	48.33	49.54	49.13	45.12	51.23	48.55	49.97	1.58
Efficiency	30	18.15	17.79	18.50	18.32	16.32	19.82	17.43	18.86	0.93

**Table 4 ijerph-18-12286-t004:** Effect of selected variables on the results of special fitness tests.

Pearson Correlation (r)	Activeness	Efficiency	Effectiveness
VO_2_peak	0.79	0.57	0.73
	*p* < 0.05	*p* < 0.05	*p* < 0.05
Evasive action test [s]	0.05	0.07	−0.04
	*p* > 0.05	*p* > 0.05	*p* > 0.05
Hip turning speed test [s]	0.18	0.19	0.25
	*p* > 0.05	*p* > 0.05	*p* > 0.05
Speed punches test [s]	0.83	0.46	0.34
	*p* < 0.05	*p* < 0.05	*p* > 0.05
1/Flexibility Ind	0.55	0.67	0.73
	*p* < 0.05	*p* < 0.05	*p* < 0.05
SJFT (Index)	0.64	0.62	0.59
	*p* < 0.05	*p* < 0.05	*p* < 0.05
SJFT (number of throws)	0.78	0.73	0.69
	*p* < 0.05	*p* < 0.05	*p* < 0.05
KSPFT	0.78	0.35	0.44
	*p* < 0.05	*p* > 0.05	*p* < 0.05

**Table 5 ijerph-18-12286-t005:** Correlation coefficient between aerobic capacity and the results of special fitness tests.

Pearson Correlation (r)	VO_2_peak
SJFT	0.36
	*p* > 0.05
KSPFT	−0.11
	*p* > 0.05
Evasive action test [s]	−0.038
	*p* > 0.05
Hip turning speed test [s]	−0.14
	*p* > 0.05
Speed punches test [s]	0.27
	*p* > 0.05

## Data Availability

The data presented in this study are available on request from the corresponding author.

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
