# Peer review of "Evaluation of the Level of Technical and Tactical Skills and Its Relationships with Aerobic Capacity and Special Fitness in Elite Ju-Jitsu Athletes"

_ijerph, 2021, doi:10.3390/ijerph182312286_

Round 1

Reviewer 1 Report

This paper explores the association between measures of aerobic capacity, “special fitness,” and jiu-jitsu performance. It is an interesting topic in an under researched population. However, it is poorly written, and the results are not well interpreted or discussed.

Abstract:

Background, first sentence: please make more concise.

Background, third sentence “Therefore, an important element seems to be the determination of the relationship between the level of special physical fitness and the technical and tactical training of athletes.”: Does not make sense. Please revise using simpler language.

Methods: Please outline the fitness tests you used.

Methods: please provide information regarding how fights were analysed.

Results: describe which fitness tests and were associated with better performance (don’t just state greater fitness).

Results: describe special efficiency indicators somewhere in methods

Conclusion: sentence reads poorly – please revise.

Introduction:

The entire introduction needs to be rewritten. Large sections of text simply do not make sense, and it is extremely difficult to read. There is too much technical jargon used without describing what those specific terms mean.

Moreover, the need for the paper is not clearly described, Try and highlight current research in the area (broadly), and the identify the gap that your paper aims to fill.

Material and Methods:

Line 83: remove the word “sports”

Line 83: change “presented” to “compete”

Line 84: provide specific inclusion and exclusion criteria

Table 1 could be removed as you have already provided this information in text.

Line 98: refer to as maximal graded treadmill test.

Line 99: define VO2max in full before using abbreviations

Line 100: need to say what equipment you used to measure oxygen uptake

Line 108: what criteria did you use to ensure that they had reached VO2max? Need to state.

Line 173: why did you space out the tests in this manner? Need to state.

Line 179: were all performance tests conducted before the tournament? If so, need to state the timeframe between performance tests and competition. If not, this needs to be addressed as a limitation as they may have changed since the date of the competition.

Please provide reliability estimates of all the tests.

Line 180: provide brand of cameras.

Line 202: provide definition of specific kicks/movements before section 2.3

Line 214: do you mean Pearson correlation coefficient? Also need to define how you interpret these results (i.e. weak, moderate, and strong), and what criteria you use to do so (i.e. Hopkins).

Results:

Line 218: should be analysis.

Line 222 – 225: simplify. This should be much more concise.

Line 235 – 245: need to state the strength of the correlations observed, and between what measures.

Discussion:

Line 247 – 252: this section does not make sense. Rewrite.

Line 253: you didn’t assess hand eye coordination, so I would remove.

Line 256 – 284: this is way too long and convoluted. Refine and make more concise. All you need to state is that the VO2max was similar to other combat sports, that it had positive associations with performance, which makes sense given the nature of the sport. Too much unnecessary background information here.

Line 288 – line 320: similar here. You have spent all this time comparing your results to other sports, which was not the purpose of the study. Simply describe which tests were associated with performance and discuss why this might be the case. This is not a literature review, but a chance to discuss the findings of your study.

Line 341: you didn’t assess flexibility, which makes this a moot point. Just talk about your results.

Conclusion:

Line 362 – 366: this doesn’t make much sense. Just state that aerobic fitness had moderate-strong associations with performance and should be prioritised during training.

Line 371 – 372: revise sentence as it does not make sense.

Author Response

Dear Reviewer,

Thank you very much for your time and valuable comments, which all have been considered and incorporated. The detailed list of responses is given below. We hope that the modifications and explanation will be acceptable for you.

Yours sincerely,

Rydzik, corresponding author

Abstract:

Background, first sentence: please make more concise.

Background, third sentence “Therefore, an important element seems to be the determination of the relationship between the level of special physical fitness and the technical and tactical training of athletes.”: Does not make sense. Please revise using simpler language.

A: This part has been corrected. The work has been corrected by a native speaker

Methods: Please outline the fitness tests you used.

A: The description has been added.

Methods: please provide information regarding how fights were analysed.

A: More information has been added

Results: describe which fitness tests and were associated with better performance (don’t just state greater fitness).

Results: describe special efficiency indicators somewhere in methods

A: This part has been corrected

Conclusion: sentence reads poorly – please revise.

A: The sentence has been rewritten

Introduction:

The entire introduction needs to be rewritten. Large sections of text simply do not make sense, and it is extremely difficult to read. There is too much technical jargon used without describing what those specific terms mean.

Moreover, the need for the paper is not clearly described, Try and highlight current research in the area (broadly), and the identify the gap that your paper aims to fill.

 A: Relevant information has been added.

Material and Methods:

Line 83: remove the word “sports”

A: This part has been corrected

Line 83: change “presented” to “compete”

A: This part has been corrected

Line 84: provide specific inclusion and exclusion criteria

A: This part has been corrected

Table 1 could be removed as you have already provided this information in text.

A: This part has been corrected

Line 98: refer to as maximal graded treadmill test.

A: This part has been corrected

Line 99: define VO2max in full before using abbreviations

A: This has been corrected and defined at the first use of the abbreviation

Line 100: need to say what equipment you used to measure oxygen uptake

A: This part has been corrected

Line 108: what criteria did you use to ensure that they had reached VO2max? Need to state.

A: This part has been complemented

Line 173: why did you space out the tests in this manner? Need to state.

A: The description has been added “The tests were supervised by the authors of the present study for 3 days, two weeks before the competition, after the completion of the direct pre-competition training period. On the first day, the aerobic capacity test (maximal graded treadmill test) was carried out. On the second day, special fitness tests (tests 1-5 given above) were performed, whereas the KSFT test was scheduled for the third day. Two days before the test, the training intensity was reduced to 30-40%.” This method of examinations was agreed with the athletes’ coaches.

Line 179: were all performance tests conducted before the tournament? If so, need to state the timeframe between performance tests and competition. If not, this needs to be addressed as a limitation as they may have changed since the date of the competition.

A: The tests were supervised by the authors of the present study for 3 days, two weeks before the competition, after the completion of the direct pre-competition training period. This method of examinations was agreed with the athletes’ coaches.

Please provide reliability estimates of all the tests.

A: All the tests used in the study have a specific validity and reliability and the methodology followed strictly the recommendations provided in the literature.

Line 180: provide brand of cameras.

A: Relevant information has been added.

Line 202: provide definition of specific kicks/movements before section 2.3

A: This part has been corrected

Line 214: do you mean Pearson correlation coefficient? Also need to define how you interpret these results (i.e. weak, moderate, and strong), and what criteria you use to do so (i.e. Hopkins).

Results:

Line 218: should be analysis.

A: This part has been corrected

Line 222 – 225: simplify. This should be much more concise.

A: This part has been corrected

Line 235 – 245: need to state the strength of the correlations observed, and between what measures.

A: This has been corrected, thank you 

Discussion:

Line 247 – 252: this section does not make sense. Rewrite.

A: This part has been corrected

Line 253: you didn’t assess hand eye coordination, so I would remove.

A: This part has been corrected

Line 256 – 284: this is way too long and convoluted. Refine and make more concise. All you need to state is that the VO2max was similar to other combat sports, that it had positive associations with performance, which makes sense given the nature of the sport. Too much unnecessary background information here.

A: This part has been corrected

Line 288 – line 320: similar here. You have spent all this time comparing your results to other sports, which was not the purpose of the study. Simply describe which tests were associated with performance and discuss why this might be the case. This is not a literature review, but a chance to discuss the findings of your study.

A: This part has been corrected

Line 341: you didn’t assess flexibility, which makes this a moot point. Just talk about your results.

A: This part has been corrected

Conclusion:

Line 362 – 366: this doesn’t make much sense. Just state that aerobic fitness had moderate-strong associations with performance and should be prioritised during training.

A: This part has been corrected

Line 371 – 372: revise sentence as it does not make sense.

A: This request has been deleted

Reviewer 2 Report

First the paper is very applicable and the fact that you used real measurements of ju-jitsu is very nice to see. The firs major correcting that needs to be made is with the tables. The formatting makes them really hard to read and understand as the lines carry over and really hard to visualize. So please make the corrections so as a reader we can quickly look at the tables and understand them. Second suggestion is to fully report the correlation data not just the correlations to activity efficiency and effectiveness. It would be nice to see the relationship between VO2 and SJFT and Speed Punches as well. Once this table is added just a brief explanation of the new correlations would be appreciated as well. Over all good work just a few visual changes and correlation data and this will be a nice study.

Author Response

(The authors gave the same response as above.)

Reviewer 3 Report

Unfortunately, this article has many serious flaws:

  • The topic of the paper is incomprehensible. The authors write about "assessment of the technical and tactical training," but in their study they did not conduct training focused on shaping technique and tactics. The authors, based on video data, analyzed the activity and efficiency of the athlete during the fight, although the description of the methodology of assessment of technique and tactics is not precise. 

  • In the introduction, the authors characterized ju-jitsu as a sport discipline, but they did not present current research results concerning the problem undertaken in the paper, did not indicate gaps in knowledge, and did not justify the rationale of the problem undertaken.
  • In selecting the subjects, the authors did not indicate clear inclusion and exclusion criteria.
  • The research methodology requires clarification. The authors did not specify what equipment they used to measure VO2max, and did not provide references to the literature for specific fitness tests.
  • The discussion section also needs serious revision.

To sum up, it is difficult to find a significant scientific problem in the article. The conclusions are quite obvious and do not move science forward.

Author Response

Dear Reviewer,

Thank you very much for your time and valuable comments, which all have been considered and incorporated. The detailed list of responses is given below. We hope that the modifications and explanation will be acceptable for you.

Yours sincerely,

Rydzik, corresponding author

Unfortunately, this article has many serious flaws:

A: It is difficult to agree with the Reviewer's comment, given the other three reviews and our knowledge of this subject area, which is presented in the manuscript. However, we have attempted to improve the manuscript using the comments made in this and other reviews. With reference to the Reviewer’s comments, we are afraid that the Reviewer did not fully understand the nature of the research. Please, review our manuscript carefully again, especially as it has been proofread to correct language issues. We hope that after the changes, the text will be acceptable and more readable to the Reviewer. The following are the detailed responses:

  • The topic of the paper is incomprehensible. The authors write about "assessment of the technical and tactical training," but in their study they did not conduct training focused on shaping technique and tactics. The authors, based on video data, analyzed the activity and efficiency of the athlete during the fight, although the description of the methodology of assessment of technique and tactics is not precise. 

A: The research was not experimental but observational in nature. The training focused on developing technical and tactical skills was controlled on a regular basis by coaches during the preparation for competitions. This is a logical procedure in the course of preparation of an athlete during the direct pre-competition training period. The analysis of video recordings of bouts and the evaluation of technical and tactical actions used and their effectiveness during the bout is a scientific novelty of this study (no one has analyzed indices of technical and tactical skills in jujitsu before), while it is an effective analysis used by coaches applied earlier in previous studies concerning combat sports. The literature review on this topic has been provided in the introduction. The description of the methodology of the evaluation of indices of technical and tactical skills has been improved to be clearer and more precise. 

  • In the introduction, the authors characterized ju-jitsu as a sport discipline, but they did not present current research results concerning the problem undertaken in the paper, did not indicate gaps in knowledge, and did not justify the rationale of the problem undertaken.

A: The introduction has been rewritten and the aim of the paper was justified.

  • In selecting the subjects, the authors did not indicate clear inclusion and exclusion criteria.

A: We have provided inclusion and exclusion criteria for the study.

  • The research methodology requires clarification. The authors did not specify what equipment they used to measure VO2max, and did not provide references to the literature for specific fitness tests.

A: The research methodology has been improved, and the description of the equipment used for VO2max measurements has been added and specified. The literature on fitness tests was provided earlier (...).

  • The discussion section also needs serious revision.

A: This section has been edited.

To sum up, it is difficult to find a significant scientific problem in the article. The conclusions are quite obvious and do not move science forward.

Reviewer 4 Report

Abstract: Minor grammatical errors/Awkward wording such as “Who presented the highest level of sport” and “fighting ju-jitsu fights” and “specificity of the rivalry characteristic of the sports variety”

Do the authors think that referring to Japanese ju-jitsu as such limits discoverability? Should there also be mention of “jujutsu” or perhaps the more familiar Brazilian spelling of jiu jitsu?

Introduction:

Awkward wording and minor grammatical errors throughout. Some examples are:

 I’m finding the first sentence to be overly wordy and difficulty to parse. It get’s better, but up until line 41 is might be able to be reworded.

Can tighten up some writing throughout by removing/lessening adjective/adverb use.

Line 49 should be something like “course of fights” or “analysis of competition” or something else, but it’s awkward as is.

Lines 49 onward get closer to explaining the background for the study aim than the line before it. There should be more emphasis on previous studies examining fitness requirements for randori, ne waza, etc. Judo and BJJ have both been studied to some extent in this regard and would likely be worth mentioning here.

Materials and methods

I’m assuming that the rendering errors for tables 1-4 are because it’s a review copy not final copy. Otherwise need to fix headings.

Might as well tell the reader the desired HR on line 104.

Author Response

Dear Reviewer,

Thank you very much for your time and valuable comments, which all have been considered and incorporated. The detailed list of responses is given below. We hope that the modifications and explanation will be acceptable for you.

Yours sincerely,

Rydzik, corresponding author

Abstract: Minor grammatical errors/Awkward wording such as “Who presented the highest level of sport” and “fighting ju-jitsu fights” and “specificity of the rivalry characteristic of the sports variety”

A: The language errors have been corrected

Do the authors think that referring to Japanese ju-jitsu as such limits discoverability? Should there also be mention of “jujutsu” or perhaps the more familiar Brazilian spelling of jiu jitsu?

A: The most common transcriptions of hand-to-hand fighting native to Japan are the KANJI characters according to Hepburn: JU-JITSU and JU-JUTSU, but the hyphen (-) connecting these characters is sometimes omitted and then you will find JUJITSU and JUJUTSU. In Spanish-speaking environments, you can find transcriptions of JIU-JITSU and JIU-JUTSU spelled separately (hyphenated) or jointly as JIUJITSTU, JIUJUTSU. Furthermore, our study is based on the athletes associated in the world JU-JITSU organizations in which this terminology is valid (according to the requirements)

Introduction:

Awkward wording and minor grammatical errors throughout. Some examples are:

 I’m finding the first sentence to be overly wordy and difficulty to parse. It get’s better, but up until line 41 is might be able to be reworded.

Can tighten up some writing throughout by removing/lessening adjective/adverb use.

Line 49 should be something like “course of fights” or “analysis of competition” or something else, but it’s awkward as is.

Lines 49 onward get closer to explaining the background for the study aim than the line before it. There should be more emphasis on previous studies examining fitness requirements for randori, ne waza, etc. Judo and BJJ have both been studied to some extent in this regard and would likely be worth mentioning here.

A: Thank you for your comment, the relevant parts have been corrected. The work has been corrected by a native speaker

Material and Methods

I’m assuming that the rendering errors for tables 1-4 are because it’s a review copy not final copy. Otherwise need to fix headings.

A: This part has been corrected

Might as well tell the reader the desired HR on line 104.

A: HR max values have been provided in Table 2. They have been provided individually for each participant.

Round 2

Reviewer 1 Report

Thank you for taking the time to address my initial comments. However, I would request the following changes be made to further strengthen the paper.

I would also ask that you go through the introduction and discussion one more time to improve the writing. Again, at times it is difficult to read.

Abstract:

Line 19: you need to state that they also need to develop good physical fitness in your background section, as this is one of your aims.

Line 23: what do you mean “seems to be essential” – I would just state your research aim here instead of this sentence.

Line 25… “were used” for what purpose? Please say.

Line 30: change “turned out to be” to “were”

Line 33: change “To achieve greater efficiency and effectiveness in the Fighting System variant of sport ju-jitsu” to something much more concise.

Introduction:

Line 53: don’t report/mention your results in the introduction.

Line 58: change “maximum” to “maximal”

Line 59: what do you mean “basic energy source.” Be more specific.

Line 60: the part in brackets should be its own sentence. Clearly state that these findings would suggest that jiu-jitsu performance requires both aerobic and anaerobic endurance.

Line 71: what do you mean “coaching control”? please explain in more detail.

Line 72: change “analyses” to “analysis”

Line 75:  Change “concerned” to “focused on” (or something similar).

Material and Methods:

Line 97: please explain what you mean by “The exclusion criterion was a negative opinion of a sports physician before the competition” be more specific.

Line 105: not “the” aerobic capacity. Should just be “aerobic capacity”

Line 115: please describe this cessation of exercise as “volitional exhaustion” to be consistent with the literature.

Line 115: In addition to volitional exhaustion, did you use any other methods to confirm that the participants truly reached VO2max (some common indices would be an RER of more than 1.1, and RPE of 19, and a HR of >90% age-predicted maximum). If you didn’t use any of these measures, you should refer to it as VO2peak throughout, as you cannot be certain that they reached VO2max.

Line 122: Please provide a sentence stating why you assessed LT2

Line 129 – 184: As per my previous comments, please provide the published reliability information of these tests in your manuscript. And please provide the relevant citations for each test (i.e. the first appearance in the published literature) within this section to support the way in which they were conducted.

Line 193: just say “maximal graded exercise test” – no need to say aerobic capacity test, and then put it in brackets.

Line 235: please provide a citation to support how you interpreted the correlations. Also, I am not sure these are correct (typically <0.2 = trivial, 0.2-0.5 = small, 0.5-0.8, moderate, >0.8 = large).

Results:

At no point do you provide results with respect to VT2. Did you include this for any purpose? If you did not, then remove it from the methods OR provide the results of this test.

Line 243: change to “Mean aerobic capacity was 51.83 ml/kg/min.” try and be more concise.

Line 261: prove the exact r value in text for your moderate correlation.

Table 4 and Table 5: please provide the specific P values for each correlation rather than just < or > 0.05

Line 282: did you attempt to perform, or did you perform?

Line 285: remove “and results in”

Line 294: no need to state that they were better/worse than other athletes. Just state that they report similar levels to other high level combat athletes.

Line 300: try and avoid writing in first person. Change “our study” to “the present study”

Line 307: “requires oxygen sources” does not make sense. Please edit this sentence using more specific and relevant language.

Line 313: remove this sentence. You did not compare test scores with Judo athletes.

Line 328: what do you mean “The quantitative and qualitative structure directly affects the course of the fight.” Remove and be more specific in your language.

Line 350: as indicated by high correlation between what? State what tests there were correlations between.

Line 371: and of this sentence you say “into its result” which doesn’t make sense. Please revise.

Line 376: Remove this sentence as you have addressed this above.

Line 382: provide the r values of these associations.

Author Response

Dear Reviewer,

Thank you very much for your time and valuable comments, which all have been considered and incorporated. The detailed list of responses is given below. We hope that the modifications and explanation will be acceptable for you.

Yours sincerely,

Rydzik, corresponding author

Thank you for taking the time to address my initial comments. However, I would request the following changes be made to further strengthen the paper.

I would also ask that you go through the introduction and discussion one more time to improve the writing. Again, at times it is difficult to read.

Abstract:

Line 19: you need to state that they also need to develop good physical fitness in your background section, as this is one of your aims.

A: This part has been corrected

Line 23: what do you mean “seems to be essential” – I would just state your research aim here instead of this sentence.

A: This part has been corrected

Line 25… “were used” for what purpose? Please say.

A: This part has been corrected

Line 30: change “turned out to be” to “were”

A: This part has been corrected

Line 33: change “To achieve greater efficiency and effectiveness in the Fighting System variant of sport ju-jitsu” to something much more concise.

 A: This part has been corrected

Introduction:

Line 53: don’t report/mention your results in the introduction.

A: This part has been corrected

Line 58: change “maximum” to “maximal”

A: This part has been corrected

Line 59: what do you mean “basic energy source.” Be more specific.

A: This part has been corrected

Line 60: the part in brackets should be its own sentence. Clearly state that these findings would suggest that jiu-jitsu performance requires both aerobic and anaerobic endurance.

A: This part has been corrected

Line 71: what do you mean “coaching control”? please explain in more detail.

A: This part has been corrected

Line 72: change “analyses” to “analysis”

A: This part has been corrected

Line 75:  Change “concerned” to “focused on” (or something similar).

A: This part has been corrected

Material and Methods:

Line 97: please explain what you mean by “The exclusion criterion was a negative opinion of a sports physician before the competition” be more specific.

A: The athletes were not admitted to the study by a physician based on negative results of the examination conducted before the participation in tournaments. Sports physicians require that the athletes have complete blood count evaluated and head EEG. The sentence has been corrected

Line 105: not “the” aerobic capacity. Should just be “aerobic capacity”

A: This part has been corrected

Line 115: please describe this cessation of exercise as “volitional exhaustion” to be consistent with the literature.

A: This part has been corrected

Line 115: In addition to volitional exhaustion, did you use any other methods to confirm that the participants truly reached VO2max (some common indices would be an RER of more than 1.1, and RPE of 19, and a HR of >90% age-predicted maximum). If you didn’t use any of these measures, you should refer to it as VO2peak throughout, as you cannot be certain that they reached VO2max.

A: Thank you very much for your suggestion, we have corrected this part.

Line 122: Please provide a sentence stating why you assessed LT2

A:  Thank you very much for your suggestion, this part have been removed

Lines 129 – 184: As per my previous comments, please provide the published reliability information of these tests in your manuscript. And please provide the relevant citations for each test (i.e. the first appearance in the published literature) within this section to support the way in which they were conducted.

A: References have been added

Line 193: just say “maximal graded exercise test” – no need to say aerobic capacity test, and then put it in brackets.

A: This part has been corrected

Line 235: please provide a citation to support how you interpreted the correlations. Also, I am not sure these are correct (typically <0.2 = trivial, 0.2-0.5 = small, 0.5-0.8, moderate, >0.8 = large).

A: References have been added

Results:

At no point do you provide results with respect to VT2. Did you include this for any purpose? If you did not, then remove it from the methods OR provide the results of this test.

A: This part has been removed

Line 243: change to “Mean aerobic capacity was 51.83 ml/kg/min.” try and be more concise.

A: This part has been corrected

Line 261: prove the exact value in text for your moderate correlation.

A: Relevant information has been added

Table 4 and Table 5: please provide the specific P values for each correlation rather than just < or > 0.05

A: This data seems clearer. However, if the Reviewer deems it necessary, we will correct it.

Line 282: did you attempt to perform, or did you perform?

A: This part has been corrected

Line 285: remove “and results in”

A: This part has been corrected

Line 294: no need to state that they were better/worse than other athletes. Just state that they report similar levels to other high level combat athletes.

A: This part has been corrected

Line 300: try and avoid writing in first person. Change “our study” to “the present study”

A: This part has been corrected

Line 307: “requires oxygen sources” does not make sense. Please edit this sentence using more specific and relevant language.

A: This part has been corrected

Line 313: remove this sentence. You did not compare test scores with Judo athletes.

A: This sentence is a reference to research in the literature. A reference item that we forgot about has been added.

Line 328: what do you mean “The quantitative and qualitative structure directly affects the course of the fight.” Remove and be more specific in your language.

A: This part has been corrected

Line 350: as indicated by high correlation between what? State what tests there were correlations between.

A: This part has been corrected

Line 371: and of this sentence you say “into its result” which doesn’t make sense. Please revise.

A: This part has been corrected

Line 376: Remove this sentence as you have addressed this above.

A: This part has been corrected

Line 382: provide the values of these associations.

A: Relevant information has been added.

Reviewer 3 Report

Major comments

Introduction:

Lines 83-88. In my opinion, the aim of the work formulated in this way does not fill any gap in the current knowledge. The authors wonder if there are correlations between special fitness and the effectiveness and activity of the competitor during the fight. I wonder if the authors also took into account the lack of any correlation between special fitness and athletic performance during the fight. It would be much more interesting to ask which manifestations of special fitness affect the athlete's effectiveness and activity in specific attack techniques.

Methods:

Line 198. You write that “analysis of bouts was performed based on digital recordings of selected tournament fights”. There is no description of the criteria used by the authors to select the fights for analysis. Did the authors analyze a single fight or several fights of a given participant? If several, is the same number of fights analyzed for all participants? Referring to the aim of the study, it is worth asking whether the effectiveness and activity during the fight (besides technique and tactics) are affected only by the athlete's special fitness, or also by the opponent's sports level. If we analyze the fight of the subject with a very weak competitor, the indicators of effectiveness and activity will be much higher than in the case of a fight with a competitor of a similar sports level. In other words, we are as effective and active as the opponent allows us.

Results:

Lines 261-171. The description of the results is unclear. Authors should use the same test names as used in the table. Instead of "better results" you should use phrases like "higher / lower values".

Lines 274-276. It is difficult to justify an attempt to look for significant correlations between VO2max and the results of special fitness tests, since they last up to 2 minutes and, as the authors write, these are anaerobic efforts.

Discussion:

The authors misinterpret the obtained results. In lines 357-359 they write: “The activeness and effectiveness of the attack significantly correlated with the speed of the attacks. This is another element that is the basis for gaining an advantage over the opponent in the first phase of the bout.” The authors found a significant positive correlation between Speed Punches Test and Activeness (r = 0.83, table 4), and between Speed Punches Test and Efficiency (r = 0.46). A positive correlation means that the respondents who needed a longer time to complete 60 punches were characterized by greater Activeness and Efficiency.  

Further (lines 364-367) the authors write: “One of the most important findings of our research is the high positive correlation of the results of special fitness tests with the activeness, effectiveness, and efficiency in the attack. For these three indices, better results were achieved by individuals who performed best in the special throwing fitness test (SJFT)…”. Also in this case, the authors found significant positive correlations between SJFT and Activeness (r = 0.64, table 4), Efficiency (r = 0.62) and Effectiveness (r = 0.59). The authors do not provide an interpretation of the SJFT Index, but it seems logical that the better score is obtained by the one who performs more throws with a lower HR increase, resulting in a lower SJFT Index value.

Conclusions:

The conclusions do not fully correspond to the aim of the study and are based on a partially flawed analysis of the results.

Minor comments

Abstract:

Lines 23-25. This whole sentence does not sound right and needs to be edited. Why are all words capitalized in some test names (eg Special Judo Fitness Test) and only the first word in others (eg Speed punches test)?

Line 31. Replace "between" with "with".

Keywords:

The keywords should not be the same as in the title. If you use different keywords, it will increase your paper’s discoverability and visibility. The keywords: “technical and tactical skills” and “ju-jitsu” are duplicated with the title.

Introduction:

Line 56. Correct the phrase "In sport in ju-jitsu".

Lines 58-59. Correct the sentence "Physical effort during combat in sport jiu-jitsu is based on submaximal and maximum exercise." Especially the word "exercise" is not appropriate here.

Line 64. [3, 9-17] In this sentence, the authors referred to as many as 10 items of literature, the vast majority of which are works by two teams: 1) Ambroży and Rydzik, 2) Franchini et al. Consideration should be given to the legitimacy of quoting all these works.

Methods:

Lines 111-112. Change “km*h-1” to “km·h-1”.

Line 116. Change HRmax to HR.

Lines 117-126. The authors unnecessarily described so many respiratory parameters since they were not included in the Results section.

Line 127. To make it easier for the reader to find more detailed information about a specific test, each description should contain a reference to a specific source and not one collective for all tests [42-45].

Table 1. Change the HRmax unit.

Author Response

Dear Reviewer,

Thank you very much for your time and valuable comments, which all have been considered and incorporated. The detailed list of responses is given below. We hope that the modifications and explanation will be acceptable for you.

Yours sincerely,

Rydzik, corresponding author

Major comments

Introduction:

Lines 83-88. In my opinion, the aim of the work formulated in this way does not fill any gap in the current knowledge. The authors wonder if there are correlations between special fitness and the effectiveness and activity of the competitor during the fight. I wonder if the authors also took into account the lack of any correlation between special fitness and athletic performance during the fight. It would be much more interesting to ask which manifestations of special fitness affect the athlete's effectiveness and activity in specific attack techniques.

A: In our study, we aimed, for the first time, to indicate the relationship between special fitness evaluated by means of well-known and recognized tests, and the efficiency and effectiveness of the athlete during a real fight.  Once the relationship is demonstrated, in further research we can determine the relationships between already proven indices and selected techniques of the attack.  The main attribute of our study was to identify which specific indices of special fitness determine an athlete's success.  The literature analysis shows that this is the first study of its kind in sport ju-jitsu.     

Methods:

Line 198. You write that “analysis of bouts was performed based on digital recordings of selected tournament fights”. There is no description of the criteria used by the authors to select the fights for analysis. Did the authors analyze a single fight or several fights of a given participant? If several, is the same number of fights analyzed for all participants? Referring to the aim of the study, it is worth asking whether the effectiveness and activity during the fight (besides technique and tactics) are affected only by the athlete's special fitness, or also by the opponent's sports level. If we analyze the fight of the subject with a very weak competitor, the indicators of effectiveness and activity will be much higher than in the case of a fight with a competitor of a similar sports level. In other words, we are as effective and active as the opponent allows us.

A: We have selected fights from three of the most prestigious tournaments of 2020. Consequently, the level of competition was high. Three bouts of each athlete were analyzed and the semi-final or final bouts were taken into consideration. The level of all tested players was similar and very high. The description in the methodology has been completed  

Results:

Lines 261-171. The description of the results is unclear. Authors should use the same test names as used in the table. Instead of "better results" you should use phrases like "higher / lower values".

A: This part has been corrected

Lines 274-276. It is difficult to justify an attempt to look for significant correlations between VO2max and the results of special fitness tests, since they last up to 2 minutes and, as the authors write, these are anaerobic efforts.

A: This correlation was added at the request of one of the reviewers. In our opinion, this is not a problem and has a logical justification. However, if the Reviewer wishes to remove it, we will do so

Discussion:

The authors misinterpret the obtained results. In lines 357-359 they write: “The activeness and effectiveness of the attack significantly correlated with the speed of the attacks. This is another element that is the basis for gaining an advantage over the opponent in the first phase of the bout.” The authors found a significant positive correlation between Speed Punches Test and Activeness (r = 0.83, table 4), and between Speed Punches Test and Efficiency (r = 0.46). A positive correlation means that the respondents who needed a longer time to complete 60 punches were characterized by greater Activeness and Efficiency.

A: Very good point, thank you, we have removed that sentence

Further (lines 364-367) the authors write: “One of the most important findings of our research is the high positive correlation of the results of special fitness tests with the activeness, effectiveness, and efficiency in the attack. For these three indices, better results were achieved by individuals who performed best in the special throwing fitness test (SJFT)…”. Also in this case, the authors found significant positive correlations between SJFT and Activeness (r = 0.64, table 4), Efficiency (r = 0.62) and Effectiveness (r = 0.59). The authors do not provide an interpretation of the SJFT Index, but it seems logical that the better score is obtained by the one who performs more throws with a lower HR increase, resulting in a lower SJFT Index value.

A: Thank you for the comment, we have corrected this part and added additional correlation value and its interpretations. 

Conclusions:

The conclusions do not fully correspond to the aim of the study and are based on a partially flawed analysis of the results.

A: This part has been corrected

Minor comments

Abstract:

Lines 23-25. This whole sentence does not sound right and needs to be edited. Why are all words capitalized in some test names (eg Special Judo Fitness Test) and only the first word in others (eg Speed punches test)?

A: This part has been corrected. The names of the tests was presented in accordance with the terminology used in the quoted literature.

Line 31. Replace "between" with "with".

A: This part has been corrected

Keywords:

The keywords should not be the same as in the title. If you use different keywords, it will increase your paper’s discoverability and visibility. The keywords: “technical and tactical skills” and “ju-jitsu” are duplicated with the title.

A: This part has been corrected

Introduction:

Line 56. Correct the phrase "In sport in ju-jitsu".

A: This part has been corrected

Lines 58-59. Correct the sentence "Physical effort during combat in sport jiu-jitsu is based on submaximal and maximum exercise." Especially the word "exercise" is not appropriate here.

Line 64. [3, 9-17] In this sentence, the authors referred to as many as 10 items of literature, the vast majority of which are works by two teams: 1) Ambroży and Rydzik, 2) Franchini et al. Consideration should be given to the legitimacy of quoting all these works.

A: This part has been corrected

Methods:

Lines 111-112. Change “km*h-1” to “km·h-1”.

A: This part has been corrected

Line 116. Change HRmax to HR.

A: This part has been corrected

Lines 117-126. The authors unnecessarily described so many respiratory parameters since they were not included in the Results section.

A: This part has been corrected

Line 127. To make it easier for the reader to find more detailed information about a specific test, each description should contain a reference to a specific source and not one collective for all tests [42-45].

A: This part has been corrected

Table 1. Change the HRmax unit.

A: This part has been corrected